# Adaptive frequency control in smart microgrid using controlled loads supported by real-time implementation

Ahmed M. Ewais[1]☯, Ahmed M. Elnoby[1]☯, Tarek Hassan Mohamed[1]*, Mohamed Metwally Mahmoud[1]☯, Yaser Qudaih[2]☯, Ammar M. Hassan[3]

**1** Department of Electrical Engineering, Faculty of Energy Engineering, Aswan University, Aswan, Egypt, **2** Higher Colleges of Technology, HCT, Abu Dhabi, UAE, **3** Arab Academy for Science, Technology and Maritime Transport, South Valley Branch, Aswan, Egypt

☯ These authors contributed equally to this work.
* tarekhie@yahoo.com

**Data Availability Statement:** All relevant data are within the paper.

**Funding:** The author(s) received no specific funding for this work.

## Abstract

The operation of the system's frequency can be strongly impacted by load change, solar irradiation, wind disturbance, and system parametric uncertainty. In this paper, the application of an adaptive controller based on a hybrid Jaya-Balloon optimizer (JBO) for frequency oscillation mitigation in a single area smart $\mu G$ system is studied. The proposed adaptive control approach is applied to control the flexible loads such as HPs and EVs by using the JBO which efficiently controls the system frequency. The suggested technique uses the power balance equation to provide a dynamic output feedback controller. The main target is to regulate the frequency and power of an islanded single area $\mu G$ powered by a PV and a diesel generator with integrations of smart bidirectional loads (HPs and EVs) that are controlled by the proposed adaptive controller in presence of electrical random loads. Moreover, the JBO is designed to minimize the effect of the system load disturbance and parameter variations. For a better assessment, the proposed controller using JBO technique is compared with two other methods which are the coefficient diagram method (CDM) and adaptive one using classical the Jaya technique. In the obtained results, the frequency deviation is found as 0.0015 Hz, which is fully acceptable and in the range of the IEEE standards. The MATLAB simulation results reveal that the suggested technique has a substantial advantage over other techniques in terms of frequency stability in the face of concurrent disturbances and parameter uncertainties. The real-time simulation tests are presented using a dSPACE DS1103 connected to another PC via QUARC pid_e data acquisition card and confirmed the MATLAB simulation results.

## Introduction

The primary problems concerning the current climatic conditions are the production of power using conventional methods, particularly from fossil fuels [1, 2]. The scholars and researchers are really concerned about paying attention to power generation by clean and

**Competing interests:** The authors have declared that no competing interests exist.

**Abbreviations:** $\mu$Gs, microgrids; CDM, coefficient diagram method; EVs, electric vehicles; HPs, heat pumps; JBO, Jaya-Balloon optimizer; LFC, load frequency control; PEVs, plug-in electric vehicles; PI, proportional integration; PV, photo-voltaic; RESs, renewable energy sources; WTG, wind turbine power generation.

renewable energy sources (RESs) such as solar (PV) and wind [3]. RESs have drawn a lot of interest since they are simple to use and economical. The intermittency of RESs prevents us from being employed as a sole source of power generation. As a result, it is common practice to deploy controllable energy storage technologies (ESTs) in conjunction with RESs [2, 4]. From the perspective of energy production, the thermal power that formerly predominated will eventually decline and the part of RESs will increasingly rise. Furthermore, due to its high absorption capacity and potential impact on the microgrid, it is vital to research the frequency control of the microgrid ($\mu$G) [5, 6].

Most of $\mu$G consists of diesel generators, RESs, ESTs, and other apparatus, and there is a power connection among these sources, which can greatly increase the $\mu$G's security and resilience [7–9]. Nonetheless, because of the more complicated topology of the $\mu$G, there are more difficulties in the system's synthesis, energy management of sources, and the structure's control and design [10–12]. RESs result in several issues such as frequency variation and change in distribution voltage due to their nature [13, 14]. Furthermore, without managing these sources properly adverse impacts occur in power systems. Therefore, effective solutions are required to maintain these characteristics to ensure the stability of the system by controlling the instantaneous power provided by RESs [15–17]. In accordance with the information in [18, 19], $\mu$Gs were described based on modeling design and communication system and were contrasted in terms of cost, reliability, and consistency.

The performance of the $\mu$G control system has been enhanced by numerous cutting-edge studies using methods including linear control, inverter regulation, and controller parameter optimization [20]. In [21], an appropriate back-to-back power converter controller was made to enhance the frequency control performance of the $\mu$G system, but numerous new elements, such as controlled loads and high-proportion new energy units results in significant design issues. Nowadays, load frequency control (LFC) has a crucial role in large-size electric power systems operation and design with complicated interconnections between its areas [22, 23]. Generally, LFC systems are designed with PI controllers. Therefore, many approaches have been discussed to adjust the gain of conventional PI controller parameters [17, 23]. Recently, the increase in variable load demands and utilization of RESs lead to system frequency fluctuations. This pushed the researchers to focus on the benefits of installing electric vehicles (EVs) and heat pumps (HPs) in $\mu$Gs as controllable loads [24–27]. In [28], a tilt-integral-derivative controller was employed with the goal of boosting the $\mu$G system's stability. A $\mu$G's LFC method built on enhanced PID was provided in [25], although it only modifies the gain of the PID controller and does not fundamentally alter the PID's control theory, reducing its adaptability to a nonlinear control system.

Additionally, it is challenging for the traditional control methods to fulfill the demands of $mu$G frequency stability in the face of increasingly highly complicated running conditions, such as stochastic power increment limitations of controllable loads in $\mu$Gs, accidental disruption of power sources and loads, alters in system structure and parameters, etc. As a result, artificial intelligence techniques are increasingly being applied in the control of $\mu$Gs to overcome the aforementioned nonlinear control difficulties. In [29], the authors proposed a dynamic programming technique with adaptable depth to the system's component, which enhanced the frequency control impact. An enhanced robust model predictive controller (MPC) with a linear quadratic regulator was used for the LFC of $mu$Gs with EVs [30]. A new LFC scheme for PV-wind-based standalone $mu$G using PID with filter—(one plus integral) cascade controller was introduced. As well as, the applied black widow optimization (BWO) was used for the first time to get the additional controller parameters. The obtained change in frequency deviation was 0.048 Hz [31]. One plus PD with a filter-fractional order PI controller and a first-ever attempt at the marine predator optimizer (MPO) helped to achieve optimal power flow

management between loads and generators. The measured frequency variation change was 0.016 Hz [32]. In [33], an innovative control method for multi-area linked power systems is the fuzzy-tilt-fractional order integral-filtered derivative controller. The controller settings are optimized using the imperialist competitive method. For the LFC of the $\mu$G system while taking into account the state of charge regulation of the battery of the EVs, [34] proposed a unique adaptive MPC technique. On the other hand, one of the algebraic robust control techniques is the coefficient diagram method (CDM) which can be used for robust control design [35, 36]. For its simplicity and reliability, CDM is considered one of the important approaches that are still used until this day. In addition, a classic optimizer called 'Jaya' [37–39], where it's proposed in this work to determine the optimal value of the integral controller due to its simplicity and speed computational time as introduced in [40, 41] to control the flexible loads (i. e. EVs and HPs) according to system dynamics.

This work suggests utilizing a Jaya-Balloon optimizer to perform adaptive frequency regulation for controllable loads in an AC smart $\mu$G. The considered $\mu$G consists of EVs, HPs, diesel generators, electrical load, and PV. The suggested new adaptive controller using hybrid Jaya+balloon optimizer is examined through the effect of frequency fluctuations resulting from both random demand loads and RESs. Furthermore, it is compared with CDM and adaptive one using Jaya techniques to show its robustness and accuracy. As well as, the real-time simulation is implemented to confirm the MATLAB simulation results. A laboratory implementation of the desired controller with the studied system is presented. In this step, the Jaya-Balloon, and Jaya algorithms of EVs and HPs are applied to real-time simulator dSPACE rt1103 and the rest of the system has been designed on PC with QUARC pid_e data acquisition card and MATLAB software with QUARC sub-program. The outputs of algorithms and the system frequency are recorded using a storage oscilloscope.

The main outstanding features of this work can be expressed as follows:

- The idea of using the controlled loads (EVs and HPs) is to compensate for the changes in power and frequency due to external disturbances and parameter uncertainties (act as a source), for normal conditions it looks like a load that absorbs power from the $\mu$G.

- The effectiveness of an integral controller adjusted by Jaya-Balloon optimizer in regulating frequency is shown in this work.

- The performance of the proposed adaptive technique is compared with that adaptive one based classical Jaya and the conventional CDM.

## Microgrid modeling and system dynamics

In this work, an islanded $\mu$G that consists of 20MW (1pu) diesel generator, 17MW (0.85pu) load, 6 MW (0.3pu) PVs, 2.38MW (0.12pu) EVs and 1.62MW (0.08pu) HPs has been suggested [19, 20]. The block diagram of LFC for the non-reheated turbine $\mu$G without a controller is shown in Fig 1.

The microgrid can be described by state-space equations as:

$$\dot{X} = AX + BU \tag{1}$$

$$Y = CX + DU \tag{2}$$

Where $A$ is the state matrix, $B$ and $D$ are input disturbance matrices, $U$ is the input disturbance vector. Also, $.X$ and $Y$ are state vector and system output, consecutively that they can be given

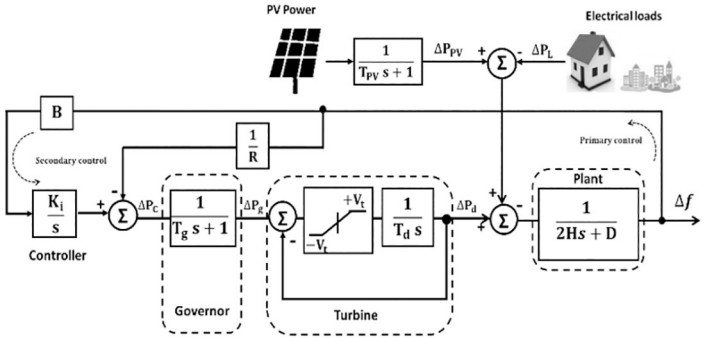

**Fig 1. Islanded single area $\mu$G model without the optimizer.**

as:

$$
\dot{X} = \begin{bmatrix} \frac{-D}{2H} & 0 & \frac{1}{2H} & \frac{1}{2H} \\ \frac{-1}{RT_g} & \frac{-1}{T_g} & 0 & 0 \\ 0 & \frac{1}{T_t} & \frac{-1}{T_t} & 0 \\ 0 & 0 & 0 & \frac{-1}{T_{PV}} \end{bmatrix} \times \begin{bmatrix} \Delta f \\ \Delta P_g \\ \Delta P_d \\ \Delta P_{PV} \end{bmatrix} +
$$

$$
\begin{bmatrix} 0 \\ \frac{-1}{T_g} \\ 0 \\ 0 \end{bmatrix} \times [\Delta P_c] + \begin{bmatrix} 0 & \frac{1}{2H} \\ 0 & 0 \\ 0 & 0 \\ \frac{1}{T_{PV}} & 0 \end{bmatrix} \times \begin{bmatrix} \Delta P_{solar} \\ \Delta P_L \end{bmatrix}
$$

(3)

$$
Y = \begin{bmatrix} 1 & 0 & 0 & 0 \end{bmatrix} \times \begin{bmatrix} \Delta f \\ \Delta P_g \\ \Delta P_d \\ \Delta P_{PV} \end{bmatrix} + [0]U
$$

(4)

where $\Delta f$, $\Delta P_g$, $\Delta P_d$, $\Delta P_L$, $\Delta P_{solar}$ and $\Delta P_c$ are the change in frequency, governor, diesel power, load power, solar power and supplementary control respectively. $D$, $H$, and $R$ are damping coefficient, inertia constant, and droop characteristics respectively. Also, $T_t$, $T_g$, $T_{PV}$ are time constants of turbine, governor and photovoltaic respectively.

## Classical Jaya algorithm

In [15], Rao introduced the standard Jaya technique. It has been classified as parameterless. Therefore, no tuning is required during computations. Jaya has additional advantages such as solving constrained and unconstrained optimization problems, being suitable for fewer design variables, and being victorious by achieving the optimal solution, which makes it more powerful. It only needs mutual control parameters such as (population size, number of design variables, and maximum number of generations). At $i^{th}$ iteration, if the best candidate gives an

optimal value of $f(x)$ in population, this means that it has become the closest to the candidate solutions and the opposite for the worst solution. The value of any $j^{th}$ variable for the $k^{th}$ candidate is $X_{j,k,i}$, which is updated based on the following equation:

$$\acute{X}_{j,k,i} = X_{j,k,i} + r_{1,k,i}(X_{j,best,i} - |X_{j,k,i}|) - r_{2,k,i}(X_{j,worst,i} - |X_{j,k,i}|) \tag{5}$$

where $\acute{X}_{j,k,i}$ the updated value of $X_{j,k,i}$; $X_{j,best,i}$ is the best value of $X_{j,k,i}$; $X_{j,worst,i}$ is the worst value of $X_{j,k,i}$, and $r_{1,k,i}$ & $r_{2,k,i}$ are random numbers between [0, 1].

$\acute{X}_{j,k,i}$ will be accepted as the required optimal solution when it gives the best function value. All accepted optimal values will be available as income to the next iteration and continue until the maximum allowable iterations are completed.

### Adaptive frequency control based classical Jaya

The unbalance between the demand for real power and its generation at an acceptable nominal frequency causes a problem in controlling the load frequency. Therefore, the proposed adaptive Jaya algorithm has been introduced into the $\mu$G to know its effect and activity in solving these issues within LFC. For the proposed $\mu$G system, PEV and HP are modeled as a first-order lag system [25, 26] as shown in Fig 2, and have been installed in residential areas for frequency regulation in the smart $\mu$G system.

Fig 3 describes the general $\mu$G block diagram with a prospective Jaya optimizer technique, where PEVi or HPi output is considered as an input to the $\mu$G. Also, Fig 4 illustrates the flowchart of Jaya algorithm.

In Fig 3, $\Delta P_L'$ represents the total loads and can be expressed as:

$$\Delta P_L' = \Delta P_L - \Delta P_{PV} - \Delta P_d + \Delta P_{cL} \tag{6}$$

where $\Delta P_{cL}$ is the participation of controlled HP and EV units, and $\Delta P_{PV}$ is the participation of photo-voltaic (PV) source.

The purpose of the system planning model is to minimize the overall cost of power generated $\Delta P_d$. For this reason, by considering ($\Delta P_L' = 0$), it is better to convert the system into 2$^{nd}$ order one for getting an objective function ($J$) with standard parameters. So, the block of ($2Hs + D$) has been added inside the proposed controller to get the standard parameter $\eta$ and $\omega_n$ as follows:

$$G(s) = \frac{\omega_n^2}{s^2 + 2\eta\omega_n s + \omega_n^2} = \frac{c \cdot ki}{T_{EV/HP}s^2 + c \cdot s + c \cdot k_i} \tag{7}$$

where $\omega_n = \sqrt{c \cdot k_i}$ and $\eta = \frac{c}{2\omega_n}$

$$M_p = e^{\frac{-\pi\eta}{\sqrt{1-\eta^2}}} = e^{\frac{-\pi\frac{c}{2\omega_n}}{\sqrt{1-\left(\frac{c}{2\omega_n}\right)^2}}} \tag{8}$$

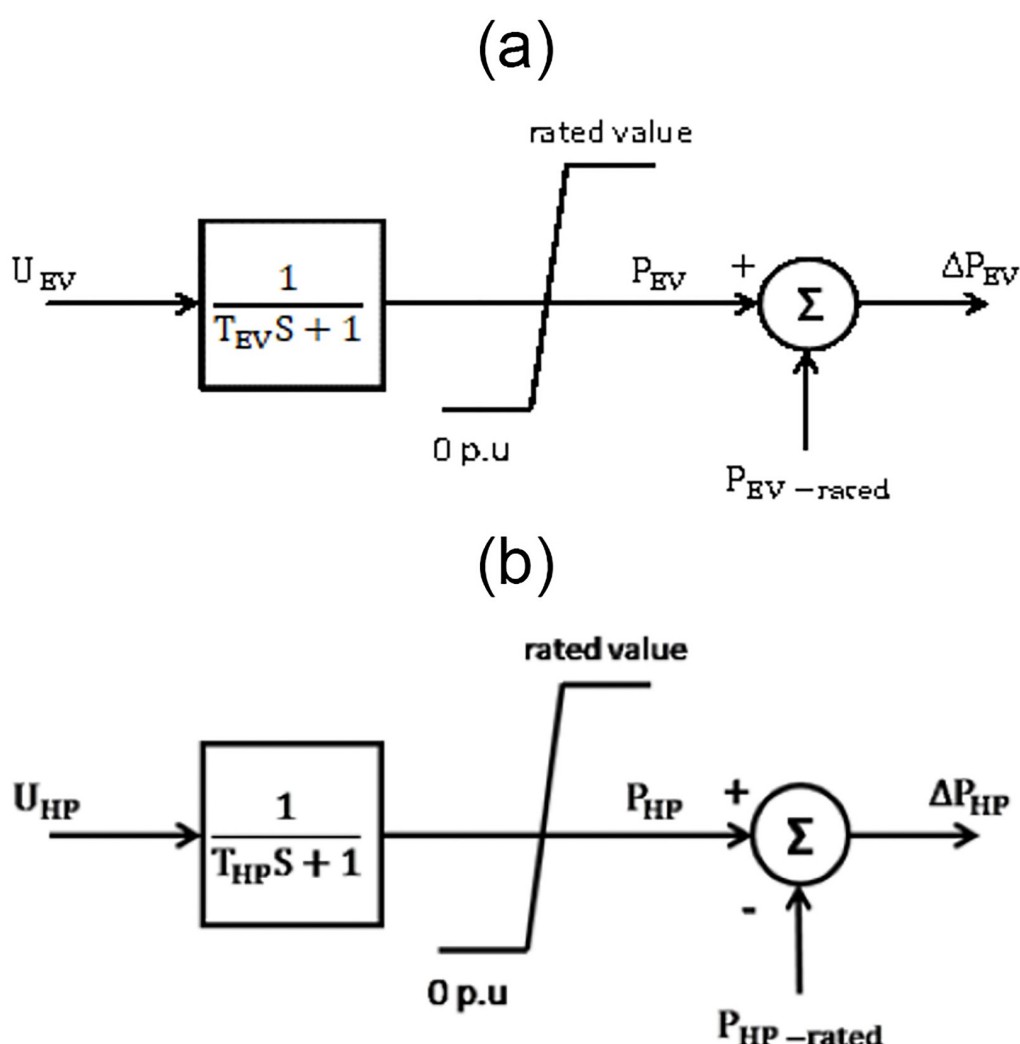

**Fig 2. Models of a) EV, b) HP.**

$$t_s = \frac{4}{\omega_n \eta} = \frac{8}{c} \tag{9}$$

$$t_r = \frac{\pi - \sqrt{1 - \eta^2}}{\omega_n \sqrt{1 - \eta^2}} = \frac{\pi - \sqrt{1 - \left(\frac{c}{2\omega_n}\right)^2}}{\omega_n \sqrt{1 - \left(\frac{c}{2\omega_n}\right)^2}} \tag{10}$$

$M_p$, $t_r$, $t_s$, $\omega_n$, and $\eta$ are the maximum overshoot, rise time, settling time, natural frequency, and damping coefficient respectively. While c is a constant value that equals 10 for HP and 3.57 for PEV. Finally, the objective function has been chosen to be:

$$J = min \sum \sqrt{t_s^2 + t_s^2 + M_p^2} \tag{11}$$

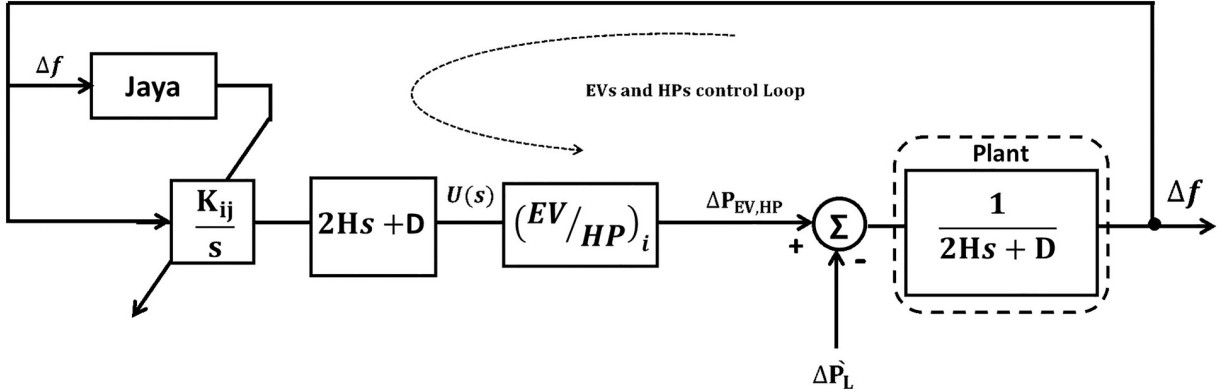

**Fig 3. General μG block diagram with optimizer.**

**Hybrid JBO method.** JBO is a modified Jaya supported by Balloon Effect (BE) identifier, the Idea of BE is to avoid the negative effect of system variations on the Jaya objective function. Fig 5, illustrates the idea of BE. As shown in Fig 4, for any iteration ($i$), $G_i(S)$ can be represented as:

$$G_i = \frac{Y_i(s)}{U_i(s)} \qquad (12)$$

Also, $G_i(S)$ can be expressed as:

$$G_i = AL_i \times G_i(s) \qquad (13)$$

where, $AL_i$ is a parameter coefficient such that:

$$G_{i-1} = \rho_i \times G_0(s) \qquad (14)$$

where,

$$\rho_i = \prod_{n=1}^{i-1} AL_n \qquad (15)$$

**Adaptive frequency control based on JBO.** The simplified block diagram of the power system using the proposed JBO for adaptive EVs and HPs control systems is shown in Fig 6.

According to the simplified model of the proposed System with JBO shown in Fig 5. It can be noted that at any iteration $i$.

$$G_i(s) = AL_i\rho_i \times G_0(s) \qquad (16)$$

where,

$$G_i(s) = AL_i\rho_i \times G_0(s) \qquad (17)$$

$$G_0(s) = \frac{1}{M_0 s + D_0} \qquad (18)$$

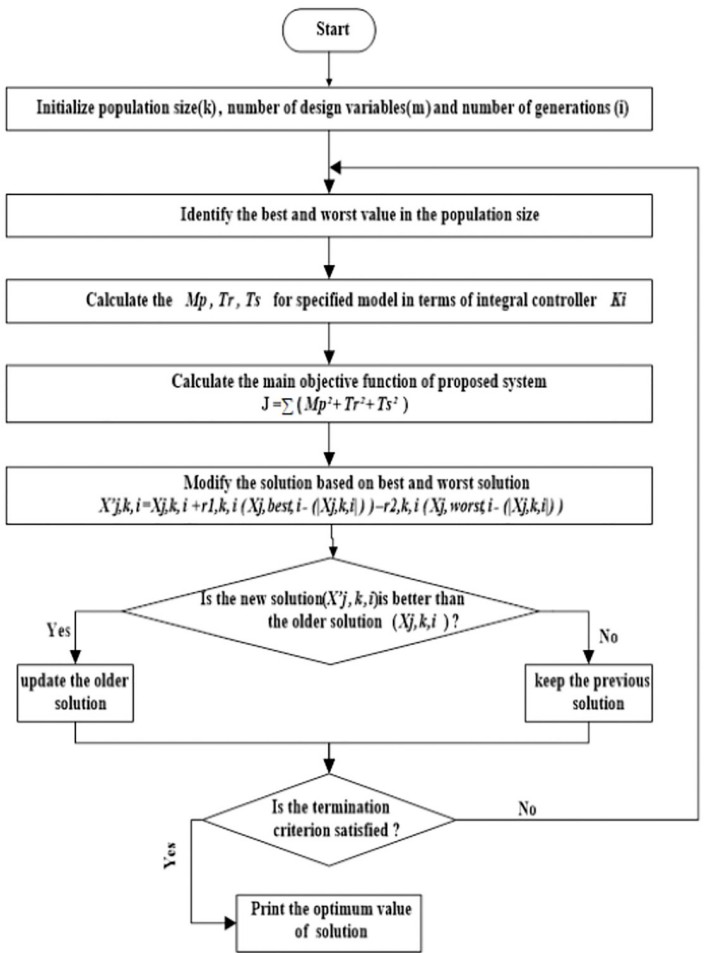

**Fig 4. Flowchart of Jaya algorithm.**

Therefore, the closed loop transfer function at any iteration ($i$) can be calculated as:

$$T.F = \frac{\left(\dfrac{k_i AL_i \rho_i}{M_0}\right)}{s^2 + \left(\dfrac{D_0 + AL_i \rho_i}{M_0}\right)s + \left(\dfrac{k_i AL_i \rho_i}{M_0}\right)} \tag{19}$$

Then

$$\omega_{n,i} = \sqrt{\left(\frac{k_i AL_i \rho_i}{M_0}\right)} \tag{20}$$

$$\eta_i = \frac{\left(\dfrac{D_0 + AL_i \rho_i}{M_0}\right)}{2\omega_{n,i}} \tag{21}$$

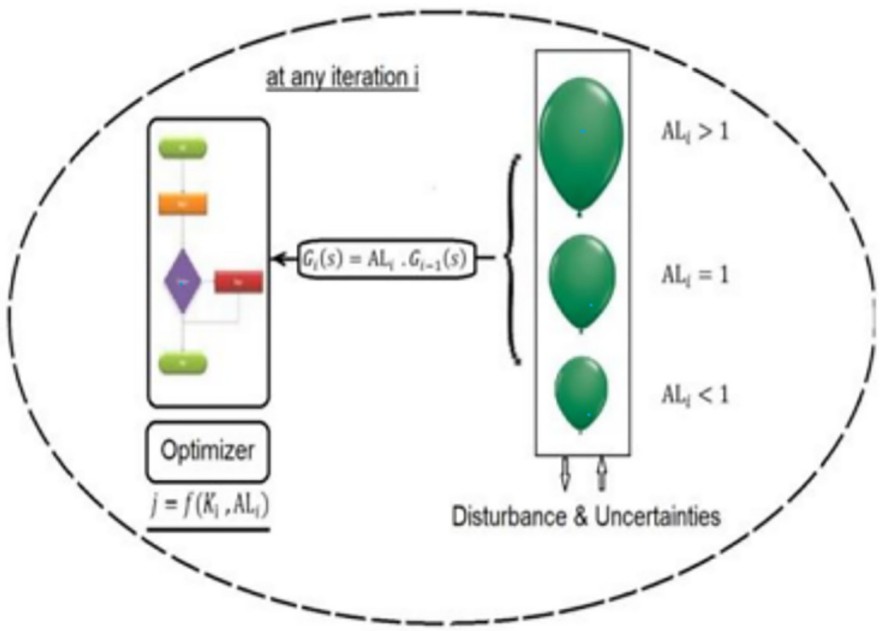

**Fig 5. Optimization strategy-based Balloon Effect identifier.**

$$M_{p,i} = e^{\left(\dfrac{-\pi(D_0 + AL_i\rho_i)}{2M_0\omega_{n,i}\sqrt{1-\eta_i^2}}\right)} \tag{22}$$

$$T_{r,i} = \frac{\pi - \sqrt{1-\eta_i^2}}{\omega_{n,i}\sqrt{1-\eta_i^2}} \tag{23}$$

$$T_{s,i} = \frac{8}{\left(\dfrac{D_0 + AL_i\rho_i}{M_0}\right)} \tag{24}$$

The objective function at any iteration $i$ can be represented as:

$$J = min \sum \left(T_{r,i}^2 + T_{s,i}^2 + M_{p,i}^2\right)^{0.5} \tag{25}$$

It is clear now that the objective function at any iteration ($i$) is a function in $k_i$, $AL_i(Obj = f(k_i, AL_i))$. This means that the system variations will affect immediately the value $AL_i$ and objective function and this will increase the ability of JBO to deal with the system difficulties. The flow chart of the Jaya algorithm is shown in Fig 7.

On the other hand, for the proposed islanded $\mu$G, many sources have been used as additional power sources besides diesel generators such as PV and power of smart flexible loads which are represented in HPs and EVs as shown in Fig 4. The dynamic relationship of generator-load between the supply error ($\Delta P_d - \Delta P_L''$) and frequency deviation ($\Delta f$) is expressed as

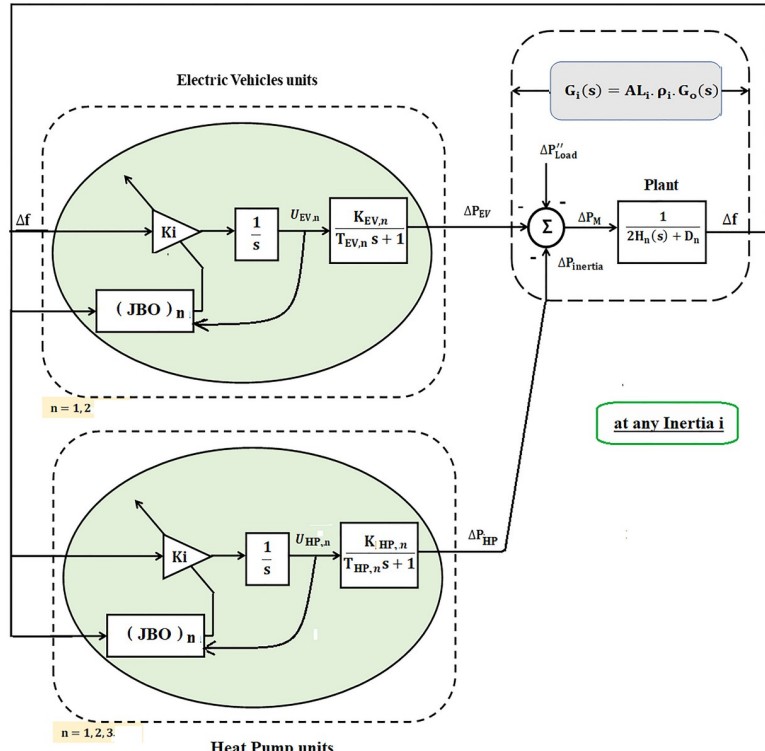

**Fig 6. Simplified microgrid model-based adaptive control system supported by BE for EVs and HPs.**

follows:

$$\dot{\Delta f} = \left(\frac{-D}{2H}\right)\Delta f + \left(\frac{1}{2H}\right)\Delta P_d + \left(\frac{1}{2H}\right)\Delta P_L'' \tag{26}$$

where

$$\Delta P_L'' = \Delta P_{EV} + \Delta P_{HP} + \Delta P_L - \Delta P_{PV} \tag{27}$$

and $(\cdot)$ denotes differential operator.

Fig 8 shows the overall islanded $\mu$G system block diagram considering participation of flexible loads (EVs and HPs) based on the adaptive Jaya optimizer.

## Results and discussions

The suggested adaptive approach is linked within the system to demonstrate the performance of LFC as shown in Fig 9. In order to approve the proposed scheme validation, digital simulations have been performed using MATLAB/Simulink software. System nominal parameters and hybrid JBO selection parameters have been proposed in this study to get the optimal value of the integral controller of flexible loads and are consecutively listed below in Tables 1 and 2.

For validation of the JBO role, contrasting three different control approaches, simulation experiments are used to examine the performance of the proposed controller under various cases. In these cases, the stability and frequency variation responses of the proposed control technique are compared to those of a designed controller built on CDM and Jaya techniques. Performance assessment of the islanded $\mu$G under the effects of RESs uncertainties and

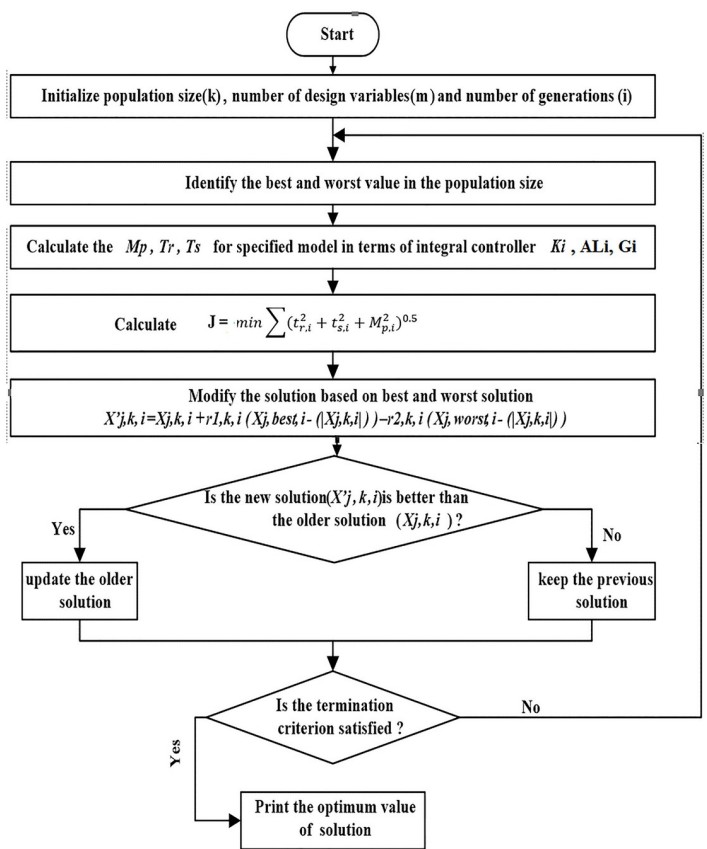

**Fig 7. Flowchart of JBO.**

random demand loads is analyzed for clarifying the role of the JBO method. The studied $\mu$G system is validated in case of random load variation and fluctuation resulting from the PV source. The simulated PV power disturbances response is shown in Fig 10(a) for 24 hr., and this PV power pattern is attained based on the incoming irradiance profile. Fig 10(b) shows a random load for 24 hr. These severe variations reflect the robustness and efficacy of the proposed adaptive controller.

The effectiveness of the investigated controllers is contrasted in Fig 11(a) in terms of frequency deviation. In terms of maximal overshoot, settling time, and steady-state error, the hybrid JBO approach outperforms Jaya and CDM methods [42]. As shown in Fig 11(a), it is noteworthy that the Jaya technique performs better and offers greater relative stability than the CDM method. Fig 11(a) indicates that the deviation of the frequency with the suggested controller is less than ± 0.0005 Hz, while this deviation arrives at ± 0.0015 Hz, and ± 0.00097 Hz in the case of CDM, and Jaya, respectively. The frequency augmentation utilizing the suggested control strategy is supported by these data. Fig 11(b) shows the deviation of diesel generator power change with the three investigated controllers. According to Fig 11(b) this deviation with the proposed controller is less than 0.00183 pu, while this deviation attains at ± 0.00432 pu, and 0.00383 pu in the case of CDM, and Jaya, respectively. The required diesel generation power using the proposed adaptive JBO is smaller than with the CDM and Jaya methods and that indicates the role of the proposed hybrid JBO. Fig 11(c) shows the deviation of EVs and HPs power change with the three studied techniques. It is clarified that; a large and fast

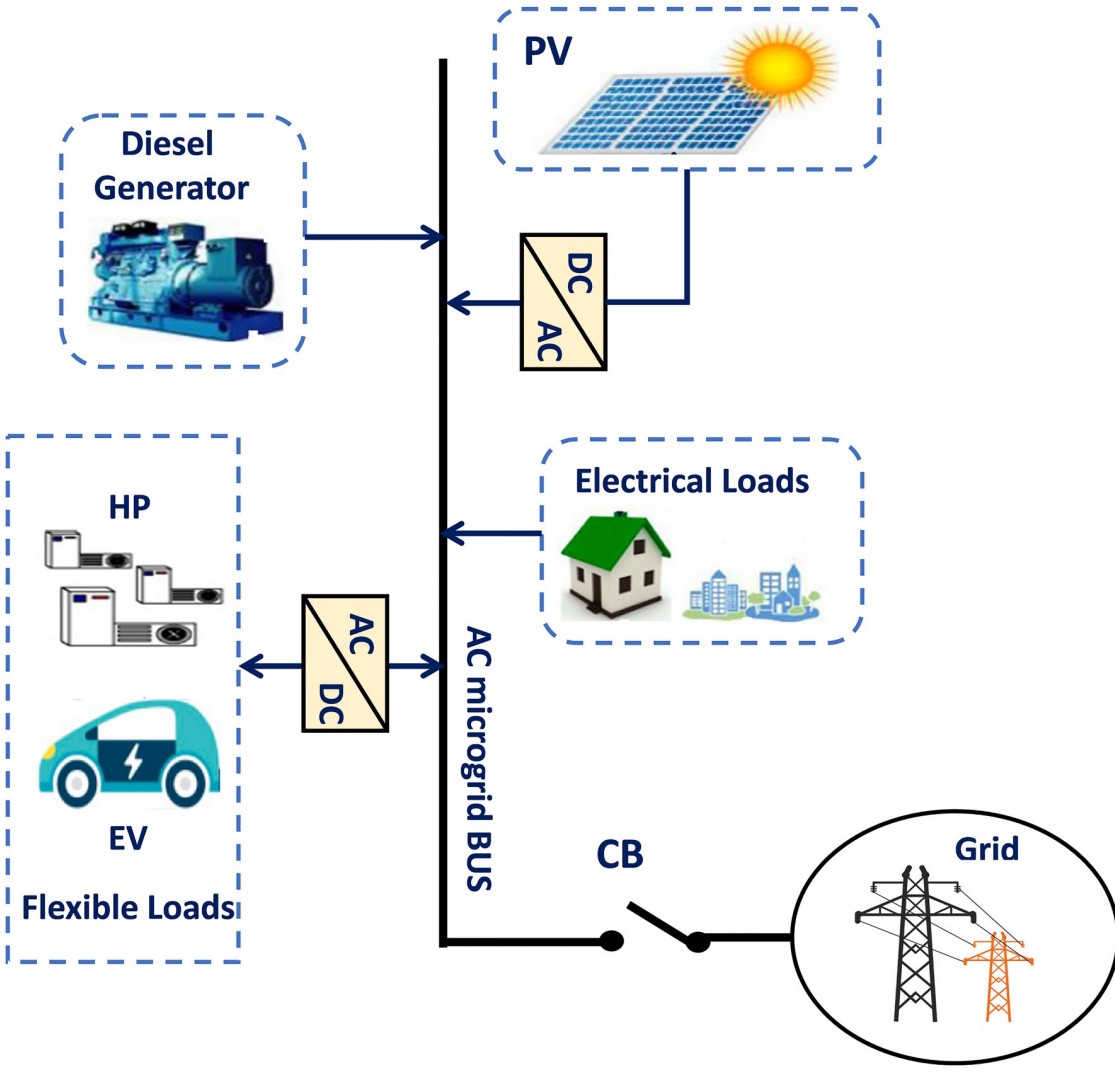

**Fig 8. Smart islanded μG system.**

discharging is taking place in the power of EVs and HPs with the proposed adaptive JBO compared with the other techniques. With the JBO the changes from (-0.008 to 0.0074), with Jaya from (-0.006 to 0.0074), and with CDM from (-0.004 to 0.002).

Fig 12, shows the implementation block diagram of the proposed system using a real-time simulator. The studied islanded μG is divided into a control setup (Jaya algorithms of HPs and EVs installed in the dSPACE rt1103 real-time simulator) and the rest of the system installed on a PC with QUARC pid_e data acquisition card (physical system). Fig 13 illustrates the physical setup of the proposed real-time simulation system. A real-time simulation test has been made at the same random load and PV source used in the previous scenario. Fig 14 shows the frequency response of the system with the CDM controller and with the proposed adaptive one. It can be noted from Fig 14, that the hybrid JBO technique can be implemented successfully to tune the controllers of the bidirectional loads such HPs and PEVs to regulate the total system frequency. This study aims to alleviate the impact of a mismatch in demand and generation on frequency, besides diminishing the variation in frequency deviation. To show and prove the

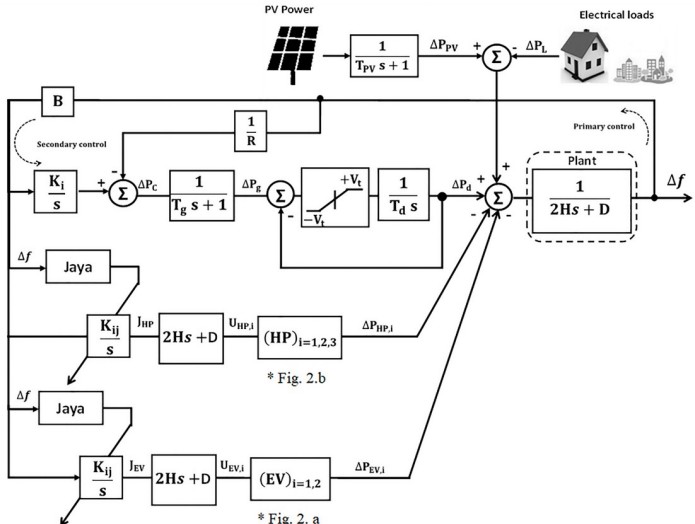

**Fig 9. Overall islanded *μ*G system block diagram considering participation of flexible loads (EVs and HPs) based on the adaptive Jaya optimizer.**

**Table 1. Data and Parameters of the Suggested *μ*G.**

| Parameter | Value | Parameter | Value |
|---|---|---|---|
| D | 0.12 | T_HP1 | 0.1 |
| M = 2H | 0.2 | T_HP2 | 0.1 |
| R | 3.0 | T_HP3 | 0.1 |
| T_g | 0.1 | T_EV1 | 0.28 |
| T_d | 0.4 | T_EV2 | 0.28 |

**Table 2. Data and Parameters Selection of Jaya Algorithm.**

| Variable | Value |
|---|---|
| Population Size (k) | 5 |
| Number of Generations (i) | 20 |
| Number of design Variables (m) | 2 |
| Upper Bound | 5 |
| Lower Bound | -5 |

effort done in this work, Table 3 is provided. This table compares the current work with previously published papers in this field in terms of simplicity, applied controller, studied cases, and *μ*G components.

## Conclusion

*μ*G components such as (EVs/HPs/PV/diesel generator/loads) are constantly changing in addition to other intermittent disturbances, which may substantially impair closed-loop performance. An adaptive controller based on hybrid JBO approach is provided for frequency

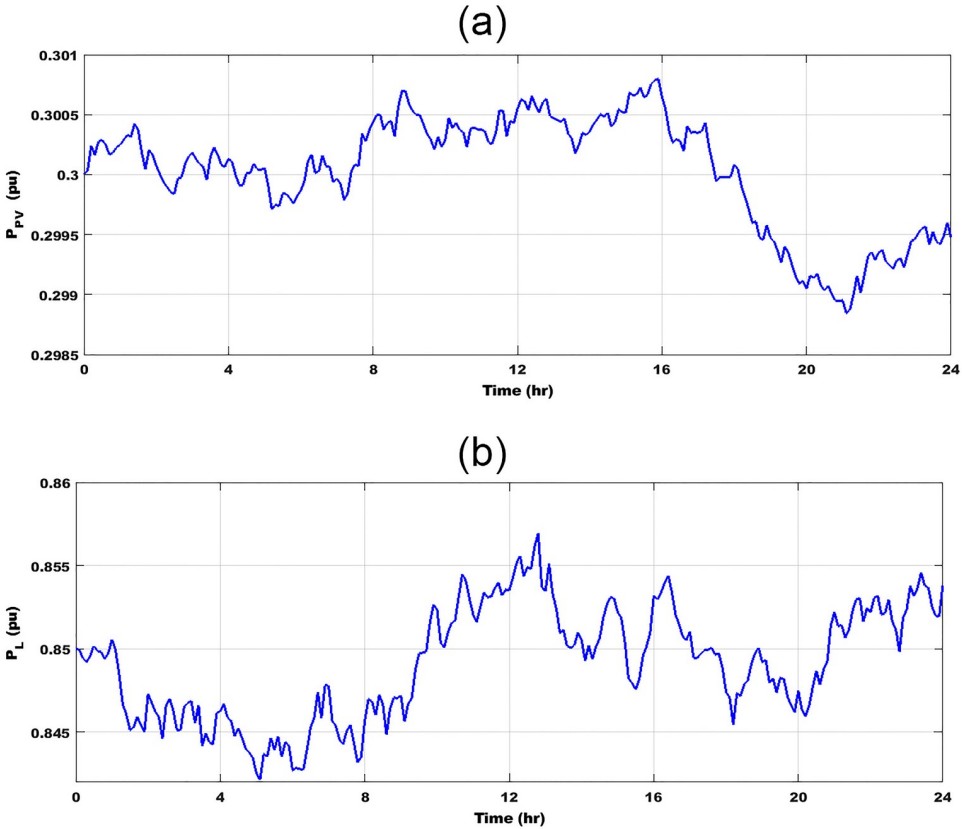

**Fig 10. Changing in (a) PV Power; (b) Random demand Load.**

regulation in the presence of various disturbances, in contrast to the overwhelming majority of classic, which are not guaranteed to deliver an acceptable performance over a wide range of running conditions. A mathematical hybrid JBO model is extracted based on $\mu$G parameters and the state-space representation of the whole system is derived. The JBO synthesis algorithm is exploited to eliminate frequency fluctuations for different uncertain conditions. The responses of frequency deviation are used to evaluate the suggested controller's performance. The simulated results show that the suggested controller performance enhancement is superior in all cases that have been considered. Digital simulation has been presented to test the system with the proposed control method under the effect of full injection of random demand loads and integration of PV sources. A comparative performance study between the proposed controller adjusted by the JBO and CDM controller has been carried out and a close analysis of the final results is obtained. It is observed that the proposed approach can effectively make online tuning of the controller gains to damp out the oscillations and provides a significant improvement within the proposed $\mu$G. Therefore, a controller with gains tuned by JBO is recommended to help in solving LFC problems and reducing oscillations inside the area. Finally, a laboratory implementation of the desired controller with the studied system was presented using dSPACE rt1103 along with QUARC pid_e data acquisition card to confirm the robustness and effectiveness of the proposed adaptive controller for PEVs and HPs on islanded $\mu$G.

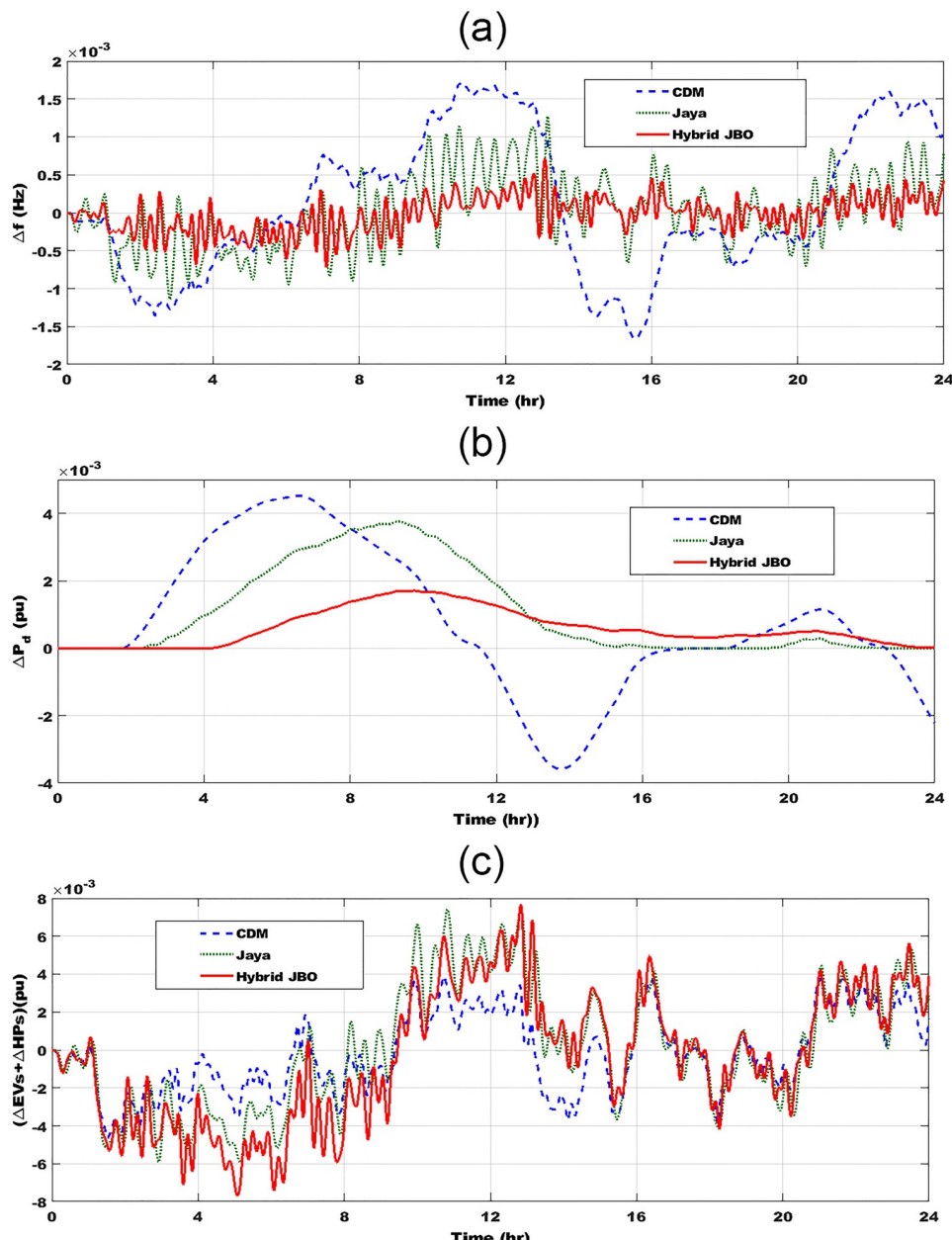

**Fig 11. System dynamic response in the first scenario: (a) frequency deviations; (b) deviation of diesel generator power change; (c) deviation of EVs and HPs power change.**

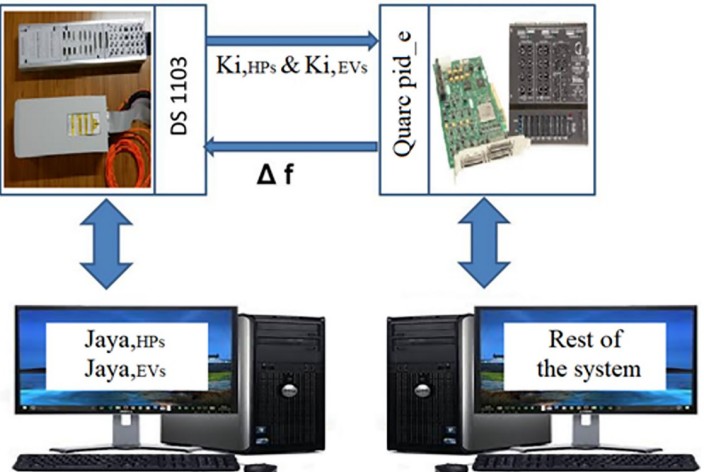

**Fig 12. Block diagram of the studied system using real-time simulation.**

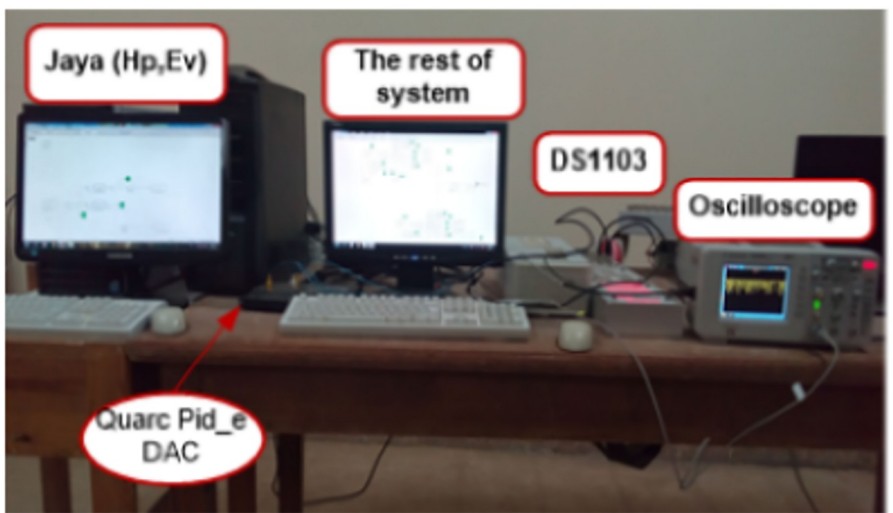

**Fig 13. Real-time laboratory setup.**

**Table 3. A comparison with previously published papers in this research area.**

| Refs. | Year | Simpli-city | $\mu G$ components | Controller | RTS | Studied cases |
|---|---|---|---|---|---|---|
| [32]] | 2022 | x | PV, WT, load, ultra-capacitor, flywheel, and diesel generator | MPO-assisted (1+PD filter+fractional order PI) | x | Constant load, variable load, stability assessment, and implementation on a real system |
| [4] | 2021 | x | PV, WT, and diesel generator | Optimized interval type-2 fuzzy logic | x | $\mu G$ in isolated mode, the effect of a short circuit in the tie line, Interrupting the power generation resources of $\mu G$ and still connected to the grid, and Islanding simultaneous with a sudden increase in the generation power of $\mu G$ |
| [31] | 2022 | x | PV, WT, load, ultra-capacitor, flywheel, and diesel generator | BWO assisted PIDF-(1 +I) | x | Constant load, a step change in load demand, controller stability, and implementation on IEEE 39 bus |
| [43] | 2021 | x | PV, load, EVs, and WTs | Optimal CDM | x | Contributions of EVs, System performance under high inertia, and high fluctuated PV and wind. |
| [42] | 2017 | ✓ | PV, WT, dump load, EVs, and diesel generator | PI | x | Control of EVs in charging or discharging power to stabilize the frequency |
| Proposed work | | ✓ | PV, load, EVs, HPs, and diesel generator | Optimized Integral | ✓ | variable load, and variable PV power |

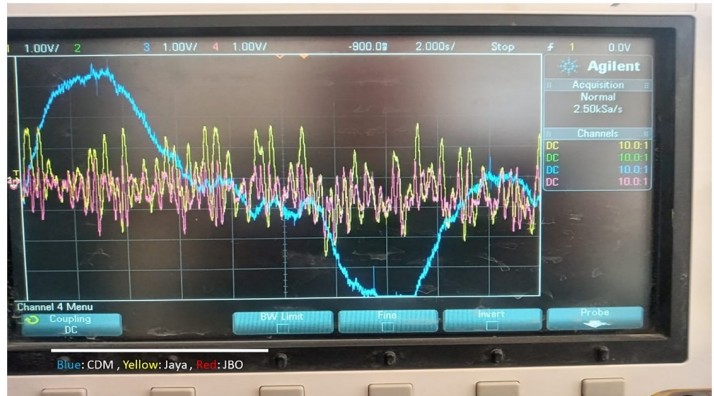

**Fig 14. System frequency deviation using real-time simulation with the three investigated controllers.**

## Author Contributions

**Conceptualization:** Tarek Hassan Mohamed.

**Data curation:** Mohamed Metwally Mahmoud.

**Methodology:** Ahmed M. Ewais, Ahmed M. Elnoby, Tarek Hassan Mohamed, Mohamed Metwally Mahmoud, Ammar M. Hassan.

**Project administration:** Tarek Hassan Mohamed, Ammar M. Hassan.

**Software:** Ahmed M. Elnoby, Ammar M. Hassan.

**Supervision:** Yaser Qudaih.

**Validation:** Ahmed M. Ewais, Ahmed M. Elnoby, Yaser Qudaih, Ammar M. Hassan.

**Writing – original draft:** Tarek Hassan Mohamed.

**Writing – review & editing:** Ahmed M. Ewais, Tarek Hassan Mohamed, Mohamed Metwally Mahmoud, Yaser Qudaih, Ammar M. Hassan.

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
