## [Decision Letter · Decision Letter 0]

5 Dec 2022

PONE-D-22-27803Adaptive frequency control in smart microgrid using controlled loads supported by real-time implementationPLOS ONE

Dear Dr. Mohamed,

Thank you for submitting your manuscript to PLOS ONE. After careful consideration, we feel that it has merit but does not fully meet PLOS ONE’s publication criteria as it currently stands. Therefore, we invite you to submit a revised version of the manuscript that addresses the points raised during the review process.

We look forward to receiving your revised manuscript.

Kind regards,

Yogendra Arya

Academic Editor

PLOS ONE

Journal Requirements:

https://journals.plos.org/pl.osone/s/file?id=wjVg/PLOSOne_formatting_sample_main_body.pdf and 

2. Our internal editors have looked over your manuscript and determined that it is within the scope of our Smart Energy Systems Call for Papers. The Collection will encompass the latest research in smart grid technologies, including information technologies, device integration, distribution methods, and data mining, all towards improving the efficiency of energy supply networks. Additional information can be found on our announcement page: https://collections.plos.org/call-for-papers/smart-energy-systems/. If you would like your manuscript to be considered for this collection, please let us know in your cover letter and we will ensure that your paper is treated as if you were responding to this call. If you would prefer to remove your manuscript from collection consideration, please specify this in the cover letter.

4. Please amend your manuscript to include your abstract after the title page.

Reviewers' comments:

Reviewer's Responses to Questions

**Comments to the Author**

1. Is the manuscript technically sound, and do the data support the conclusions?

Reviewer #1: Partly

Reviewer #2: Partly

Reviewer #3: Partly

Reviewer #4: Yes

2. Has the statistical analysis been performed appropriately and rigorously? 

Reviewer #1: N/A

Reviewer #2: N/A

Reviewer #3: Yes

Reviewer #4: Yes

3. Have the authors made all data underlying the findings in their manuscript fully available?

Reviewer #1: No

Reviewer #2: Yes

Reviewer #3: Yes

Reviewer #4: Yes

4. Is the manuscript presented in an intelligible fashion and written in standard English?

Reviewer #1: No

Reviewer #2: Yes

Reviewer #3: Yes

Reviewer #4: Yes

5. Review Comments to the Author

Reviewer #1: Paper needs to update with following comments.

1. As it is known that PV system has no inertia, so how it helps in Frequency control is a big question. Authors have to address this point.

2. There are many soft computing approaches then why Jaya Algorithm is employed for optimization?

3. Only integrating controller is used in the proposed controller. How this is possible to react in real sense without P controller.

4. There are many typos and grammar error. Reference list is not suffice to know the research gap. Some papers are self cited.

5. It should have some latest works with fractional order controllers to cover in discussion.

Reviewer #2: 1. The author claims the proposed controller is adaptive and how the JAYA optimization tuned Integral controller becomes an Adaptive controller. Proper justification is required, which should be incorporated in the Simulation Results section.

2. Why did the author choose Jaya optimization for any specific reason? The superiority is compared with the algebraic approach, so why not with any recent Metaheuristic techniques?

3. Enhance the literature review section regarding recent LFC contributions.

Reviewer #3: These are the comments to enhance the quality of work:

1) In Abstract provide some numerical data, what authors have achieved.

2) Introduction Section is weak, more explanation is required.

3) Elaborate literature review section and incorporate recent references as:

https://doi.org/10.1115/1.4056135;
https://doi.org/10.1016/j.isatra.2022.06.010; 10.1109/ACCESS.2022.3202907; https://link.springer.com/article/10.1007/s12652-021-03403-6.

4) More explanation is required for the proposed optimization scheme.

5) Rewrite conclusion section.

Reviewer #4: The paper proposes a classic Jaya optimizer for tuning the gains of frequency controllers of HP and PEV for islanded single area microgrid to improve system frequency stability. I have carefully assessed the manuscript and think that the paper has good contribution to the literature. I have the following comments: -

1) How the uncertainty owing to EVs has been considered while designing the proposed controller?

2) As there are several similar algorithms which can provide the similar results. Why shall we choose the Jaya optimization algorithm, instead of choosing "conventional" optimization solvers such as the Interior-point algorithm and other solvers?

3) Could the authors comment on what is the difference between the proposed optimization algorithm and the following paper algorithm “A new optimal robust controller for frequency stability of interconnected hybrid microgrids considering non-inertia sources and uncertainties,” International Journal of Electrical Power & Energy Systems, vol. 128. Elsevier, p. 106651, Jun. 2021. doi: 10.1016/j.ijepes.2020.106651.

4) What criteria have been considered, which led to the mentioned scenario for the test system? Especially the improvement of the frequency with the proposed controller is slightly noticed. See Fig. 7(a). Why we didn’t see sever case change (system parameter variations).

6. PLOS authors have the option to publish the peer review history of their article (what does this mean?). If published, this will include your full peer review and any attached files.

Reviewer #1: No

Reviewer #2: No

Reviewer #3: **Yes: **Dr. Pawan Kumar Pathak

Reviewer #4: No

---

## [Author Response · Author response to Decision Letter 0]

19 Jan 2023

Dear Editors and Reviewers

The authors are thankful to the learned Editor and Reviewers for their thoughtful and detailed comments to improve the quality of the manuscript. The authors have given reviewer comments a lot of interest in the revision process in an attempt to address all of the reviewers’ concerns and corrections as you will already find them incorporated in the revised manuscript. Moreover, a reply to each of the reviewers’ comments is provided below.

Kindly find the response to the reviewer’s comments in the following paragraphs. We hope this revised version of the manuscript meets the editor and reviewers’ expectations, and the standards of publication in the PLOS ONE Journal. 

The changes carried out by the authors are incorporated in the revised manuscript and highlighted in blue.

Editor's Comments:

Comments to the Authors:

Comment-1: Thank you for submitting your manuscript to PLOS ONE. After careful consideration, we feel that it has merit but does not fully meet PLOS ONE’s publication criteria as it currently stands. Therefore, we invite you to submit a revised version of the manuscript that addresses the points raised during the review process. Please submit your revised manuscript by Jan 19 2023 11:59PM. 

Response-1: Our sincere thanks and appreciation to the editor for considering our manuscript for publication in PLOS ONE Journal, and the recommending submission of the revised manuscript. To improve the quality of the manuscript, the reviewer's queries are addressed and their suggestions are incorporated into the revised manuscript. A new method called Hybrid Jaya-Balloon Optimizer is designed and implemented and compared with CDM and Jaya for frequency stability. The introduction section is rewritten, many changes in ;/abstract, simulation results, and conclusions are done based on reviewers’ quires. Table 3 is added to compare the current work with previously published works. Some sentences have been edited in the revised paper to clarify the paper's contributions and enhance the paper quality.

Comment-2: Please include the following items when submitting your revised manuscript:

Response-2: Our sincere thanks and appreciation to the editor for his comment. The required items are attached during submission process. The changes carried out by the authors are incorporated in the revised manuscript and highlighted in blue to be easily viewed by the editors and reviewers. A cover letter is provided and prepared to explain, point by point, the details of the revisions to the manuscript.

Comment-3: Please ensure that your manuscript meets PLOS ONE's style requirements.

Response-3: The authors are extremely thankful to the editor for this thoughtful point. The revised manuscript meets PLOS ONE's style.

Comment-4: Our internal editors have looked over your manuscript and determined that it is within the scope of our Smart Energy Systems Call for Papers. The Collection will encompass the latest research in smart grid technologies, including information technologies, device integration, distribution methods, and data mining, all towards improving the efficiency of energy supply networks. Additional information can be found on our announcement page: https://collections.plos.org/call-for-papers/smart-energy-systems/. If you would like your manuscript to be considered for this collection, please let us know in your cover letter and we will ensure that your paper is treated as if you were responding to this call. If you would prefer to remove your manuscript from collection consideration, please specify this in the cover letter.

Response-4: The authors are extremely thankful to the editor for his interest in our work. 

Comment-5: 3. Please note that PLOS ONE has specific guidelines on code sharing for submissions in which author-generated code underpins the findings in the manuscript. In these cases, all author-generated code must be made available without restrictions upon publication of the work. Please review our guidelines at https://journals.plos.org/plosone/s/materials-and-software-sharing#loc-sharing-code and ensure that your code is shared in a way that follows best practice and facilitates reproducibility and reuse.

Response-5: The authors are extremely thankful to the editor for his advice. The guidelines have been reviewed.

Comment-6: Please amend your manuscript to include your abstract after the title page.

Response-6: The authors are extremely thankful to the editor for his careful in improving the quality of the manuscript. The revised manuscript has been amended to include the abstract after the title page. 

Reviewers Comments:

Reviewer 1 

Comments to the Authors:

Comment-1: As it is known that PV system has no inertia, so how it helps in Frequency control is a big question. Authors have to address this point.

Response-1: At the beginning, the authors are thankful to the honorable reviewer for the words of encouragement and trust in our work. The PV has no inertia and cause problem due to its intermittent nature. In our study the inertia is obtained from storage units (EVs, and HPs). We applied a newly hybrid Jaya-Balloon optimizer (JBO)-based controller for frequency oscillation mitigation in the investigated smart µG. The proposed optimization is compared with the conventional CDM and Jaya. Kindly check the revised manuscript.

Comment-2: There are many soft computing approaches then why Jaya Algorithm is employed for optimization?

Response-2: The authors are thankful to the esteemed reviewer upon his valuable comment. Based on the reviewer comment, the authors applied a newly hybrid Jaya-Balloon optimizer (JBO) and compared it with robust techniques such as Jaya and CDM. Kindly, check pages No. (8 - 12).

Comment-3: Only integrating controller is used in the proposed controller. How this is possible to react in real sense without P controller.

Response-3: The authors are thankful to the valuable comment, but we want to shed the light on this point, Integral controller is a suitable classical one in case of two main reasons:

1- The simplified closed loop transfer function of the LFC system is second order one, so including integral controller will make a desired output performance is available.

2- According to physical characteristics of the governor, the changes in its inputs should be in slow action, so integral controller is suitable one in such case.

3- Most references in LFC use integral classical controller.

Comment-4: There are many typos and grammar error. Reference list is not suffice to know the research gap. Some papers are self-cited.

Response-4: The authors are thankful to the esteemed reviewer upon his valuable comment and apologize for these typos and grammar errors. The revised paper has been proof read and grammatical errors corrected. Some references have been added to strength the literature review and most of self-cited papers are removed. Furthermore Table 3 is presented and added in the revised paper to show the current work importance and novelty. Kindly, check pages No. (19 - 23). 

Comment-5: It should have some latest works with fractional order controllers to cover in discussion.

Response-5: The authors are thankful to the esteemed reviewer upon his valuable comment. Some new references regarding latest works with fractional order controllers have been added to the literature part in Section 1 (Introduction). Furthermore, Table 3 is provided in the revised paper to strength the literature review. Kindly, check pages No. (2, 3, and 19).

Reviewer 2 

Comment-1: The author claims the proposed controller is adaptive and how the JAYA optimization tuned Integral controller becomes an Adaptive controller. Proper justification is required, which should be incorporated in the Simulation Results section.

Response-1: The authors are thankful to the esteemed reviewer upon his valuable comment. According to the reviewer’s viewpoint, more description about using Jaya as adaptive controller has been added, also a flowchart of Jaya in adaptive frequency controller has been considered in the revised manuscript. (Figures 4, 6 and 7)

Comment-2: Enhance the literature review section regarding recent LFC contributions.

Response-2: The authors are thankful to the esteemed reviewer upon his valuable comment. The literature review section regarding recent LFC contributions is enhanced in the revised paper. The introduction part is re written to be improve the paper quality and readability based on the reviewer comment. There are some references have been added to the literature part in Section 1 (Introduction) to strength the literature review. Furthermore, a comparative with previously published articles in this research area is provided in Table 3. Kindly, check pages No. (2, 3, 4, and 19). 

Comment-3: Why did the author choose Jaya optimization for any specific reason? The superiority is compared with the algebraic approach, so why not with any recent Metaheuristic techniques?

Response-3: The authors are extremely thankful to the reviewer for this thoughtful point. As suggested by the respected reviewer, we applied a newly hybrid Jaya-Balloon optimizer (JBO)-based controller for mitigating the effect of mismatch in demand and generation and minimize the change in frequency deviation in the investigated smart µG. The main benefit of this technique is its high effectiveness, small overshoot, and quick dynamic response. The proposed optimization is compared with the conventional CDM and Jaya. Results section presents and summarize a comparison of the results between the proposed and the conventional CDM and Jaya. Kindly, check pages No. (8 - 12). 

Reviewer 3

Comment-1: In Abstract provide some numerical data, what authors have achieved.

Response-1: As suggested by the esteemed reviewer, the abstract is rewritten and involve a numerical data. Kindly check (Pages no. 1). 

Comment-2: Introduction Section is weak; more explanation is required.

Response-2: The authors are thankful to the esteemed reviewer upon his valuable comment. The introduction part is re written to be improve the paper quality and readability based on the reviewer comment. Some paragraphs have been added. There are some references have been added to the literature part in Section 1 (Introduction) to strength the literature review. Furthermore, a comparative with previously published articles in this research area is provided in Table 3. Kindly, check pages No. (1, 2, 3, and 4). 

Comment-3: Elaborate literature review section and incorporate recent references as:

https://doi.org/10.1115/1.4056135;
https://doi.org/10.1016/j.isatra.2022.06.010; 10.1109/ACCESS.2022.3202907; https://link.springer.com/article/10.1007/s12652-021-03403-6.

Response-3: Thank you for this comment. The suggested papers have been cited and they were helpful. The literature review section is elaborated in the revised paper. Kindly check (Pages no. 20-23).

Comment-4: More explanation is required for the proposed optimization scheme.

Response-4: We applied a newly hybrid Jaya-Balloon optimizer (JBO)-based controller for frequency oscillation mitigation in the investigated smart µG. The proposed optimization is compared with the conventional CDM and Jaya. As suggested by the esteemed reviewer, a more explanation is added for the proposed optimization scheme. Kindly check (Pages no. 6 - 12).

Comment-5: Rewrite conclusion section.

Response-5: The authors are extremely thankful to the reviewer for this thoughtful point. The conclusion section is rewritten to include the numerical results of the proposed work based on the esteemed reviewer suggestion. Kindly check (Pages no. 19).

Reviewer 4

Comment-1: How the uncertainty owing to EVs has been considered while designing the proposed controller?

Response-1: The authors are thankful to the esteemed reviewer upon his valuable comment. We want to shed the light on this point: Robustness and stability of the Jaya + BE scheme has been explained be Ammar M. Hassan and Tarek Hassan Mohamed in “A novel adaptive load frequency control in single and interconnected power systems” Ain Shams Engineering Journal

Volume 12, Issue 2, June 2021, Pages 1763-1773, https://doi.org/10.1016/j.asej.2020.08.024, so the uncertainty owing to EV’s and HPs has been considered while designing the proposed controller.

Comment-2: As there are several similar algorithms which can provide the similar results. Why shall we choose the Jaya optimization algorithm, instead of choosing "conventional" optimization solvers such as the Interior-point algorithm and other solvers?

Response-2: The authors are extremely thankful to the reviewer for this thoughtful point. A newly hybrid Jaya-Balloon optimizer (JBO)-based controller for frequency oscillation mitigation in the investigated smart µG is applied and compared with conventional techniques. The new methods prove its superiority compared with conventional optimization solvers as provided in the literature. Kindly check (Pages no. 20 - 23).

Comment-3: Could the authors comment on what is the difference between the proposed optimization algorithm and the following paper algorithm “A new optimal robust controller for frequency stability of interconnected hybrid microgrids considering non-inertia sources and uncertainties,” International Journal of Electrical Power & Energy Systems, vol. 128. Elsevier, p. 106651, Jun. 2021. doi: 10.1016/j.ijepes.2020.106651.

Response-3: The authors are extremely thankful to the reviewer for this thoughtful point. The main difference between our manuscript and the mentioned paper can be concluded as follow: 

The mentioned paper used only an optimal CDM controller to control the virtual inertia connected to power system, but we used 3 controllers: 1- optimal CDM, 2- Adaptive controller based Jaya optimization and 3- Adaptive controller based JBO to control both EVs and HPs units connected to the power system.

Comment-4: What criteria have been considered, which led to the mentioned scenario for the test system? Especially the improvement of the frequency with the proposed controller is slightly noticed. See Fig. 7(a). Why we didn’t see sever case change (system parameter variations).

Response-4: The authors are extremely thankful to the reviewer for this thoughtful point. For our under study system, the components are small size (diesel generator, turbine and governor). In such case sever parameters variations are not practical and the load disturbance is the effective system difficulty. 

The authors once again thank the learned Editors and Reviewers for their valuable comments for improving the quality of the manuscript.

---

## [Decision Letter · Decision Letter 1]

13 Mar 2023

Adaptive frequency control in smart microgrid using controlled loads supported by real-time implementation

PONE-D-22-27803R1

Dear Dr. Mohamed,

We’re pleased to inform you that your manuscript has been judged scientifically suitable for publication and will be formally accepted for publication once it meets all outstanding technical requirements.

Kind regards,

Yogendra Arya

Academic Editor

PLOS ONE

Additional Editor Comments (optional):

Reviewers' comments:

Reviewer's Responses to Questions

**Comments to the Author**

1. If the authors have adequately addressed your comments raised in a previous round of review and you feel that this manuscript is now acceptable for publication, you may indicate that here to bypass the “Comments to the Author” section, enter your conflict of interest statement in the “Confidential to Editor” section, and submit your "Accept" recommendation.

Reviewer #1: All comments have been addressed

Reviewer #3: All comments have been addressed

2. Is the manuscript technically sound, and do the data support the conclusions?

Reviewer #1: Yes

Reviewer #3: Yes

3. Has the statistical analysis been performed appropriately and rigorously? 

Reviewer #1: Yes

Reviewer #3: Yes

4. Have the authors made all data underlying the findings in their manuscript fully available?

Reviewer #1: No

Reviewer #3: Yes

5. Is the manuscript presented in an intelligible fashion and written in standard English?

Reviewer #1: Yes

Reviewer #3: Yes

6. Review Comments to the Author

Reviewer #1: No further comment on the revised manuscript. The manuscript can be accepted if all comments are address by other reviewer together.

No further comment on the revised manuscript. The manuscript can be accepted if all comments are address by other reviewer together.

Reviewer #3: (No Response)

7. PLOS authors have the option to publish the peer review history of their article (what does this mean?). If published, this will include your full peer review and any attached files.

Reviewer #1: **Yes: **Utkal Mehta, University of the South Pacific (USP), Fiji

Reviewer #3: **Yes: **Dr. Pawan Kumar Pathak

---

## [Editor Report · Acceptance letter]

30 Mar 2023

PONE-D-22-27803R1 

Adaptive frequency control in smart microgrid using controlled loads supported by real-time implementation 

Dear Dr. Mohamed:

I'm pleased to inform you that your manuscript has been deemed suitable for publication in PLOS ONE. Congratulations! Your manuscript is now with our production department. 

Kind regards, 

on behalf of

Dr. Yogendra Arya 

Academic Editor

PLOS ONE